# Comparison of Point Cloud Registration Algorithms for Mixed-Reality Cross-Device Global Localization

Alexander Osipov [1,*], Mikhail Ostanin [1] and Alexandr Klimchik [2]

[1] Institute of Robotics and Computer Vision, Innopolis University, 420500 Innopolis, Russia
[2] School of Computer Science, University of Lincoln, Lincoln LN6 7TS, UK
* Correspondence: a.osipov@innopolis.university

**Abstract:** State-of-the-art approaches for localization and mapping are based on local features in images. Along with these features, modern augmented and mixed-reality devices enable building a mesh of the surrounding space. Using this mesh map, we can solve the problem of cross-device localization. This approach is independent of the type of feature descriptors and SLAM used onboard the AR/MR device. The mesh could be reduced to the point cloud that only takes vertices. We analyzed and compared different point cloud registration methods applicable to the problem. In addition, we proposed a new pipeline Feature Inliers Graph Registration Approach (FIGRA) for the co-localization of AR/MR devices using point clouds. The comparative analysis of Go-ICP, Bayesian-ICP, FGR, Teaser++, and FIGRA shows that feature-based methods are more robust and faster than ICP-based methods. Through an in-depth comparison of the feature-based methods with the usual fast point feature histogram and the new weighted height image descriptor, we found that FIGRA has a better performance due to its effective graph-theoretic base. The proposed pipeline allows one to match point clouds in complex real scenarios with low overlap and sparse point density.

**Keywords:** indoor collaborative localization; augmented and mixed-reality devices; point cloud registration; comparison

## 1. Introduction

Localization and mapping are the main technical capabilities of modern mixed-reality and robotics systems. This functionality allows the combining of the real with the digital worlds into a single reality. A single coordinate space is required for people to cooperate in mixed reality, or for people and robots to work together. In addition, device localization is required to place content on a pre-built map. The collaborative localization creates wide opportunities to develop programs with a spatial multi-user experience. A good case in point is creating games with cooperative interaction [1,2], as well as collaborative modeling and design [3,4] in the construction industry [5–7], presenting and digital content in education [8–10], and in tourism [11].

Various devices perform localization and mapping on board. However, there is a trend towards cloud computing and moving content localization to the cloud. For example, some systems are based on anchors such as Microsoft Azure Spatial Anchors [12], Niantic [13], Google Cloud Anchors [14], etc. These systems send key points and descriptors to the cloud for content localization. Certain solutions use visual landmarks for spatial anchors (data matrices, hand tracking) [15]. Such solutions have limitations because they require constant landmark visibility and calibration. It should be noted that anchor-based solutions suffer from spatial accuracy divergence in the distant-anchor regions.

Modern on-board systems have several limitations in terms of co-localization. First, if devices use different features, this complicates their co-localization [16]. In addition, various systems use different simultaneous localization and mapping (SLAM) algorithms and have hardware acceleration for them. For example, various features can be used,

such as SIFT [17] or SOSNet [18]. Moreover, existing devices will lag behind the new SLAM algorithms. This lag makes it impossible for the systems to co-localize with each other. In addition to the above, a human finds it difficult or impossible to perceive a map consisting of features. This limitation imposes a restriction on the remote content installation on the map.

A common functionality allows constructing a mesh map of the real environment in mixed-reality systems. The mesh map should geometrically represent the real environment. These maps should be similar to one another in the same place on different systems regardless of the algorithms. However, the mesh itself has a lot of information which is difficult to transfer and save, so we propose to reduce the spatial map to a point cloud consisting only of the mesh vertices. The result is a sparse cloud of points that is easier to save and transfer. At the same time, this point cloud will constitute human-readable perception, which could help in using it for content placement. Moreover, this approach can be scaled to the co-localization of any devices, also building a 3D mesh map. For example, this common functionality is also adapted to android devices that have at least one camera sensor on board in Niantic's Lightship ARDK [13].

In this paper, we propose the registration pipeline Feature Inliers Graph Registration Approach (FIGRA). This paper was motivated by developing an approach that is able to solve the problem of indoor mixed-reality cross-device localization. We collected a new dataset with pairs of reconstructed point clouds with mixed-reality devices (Microsoft HoloLens first and second generation) for one environment and analyzed point cloud registration methods with modifications.

At the first stage, we experimentally compared the efficiency of four global registration methods and pipeline FIGRA on the real point clouds of indoor environments. This efficiency comparison was needed to learn which methods are more suitable for solving the problem of MR devices co-localization, as well as to understand which parameters affect the algorithm robustness and registration success. At the second stage, we minutely analyzed the efficiency of most perspective registration methods with different parameters and method modifications. As a modification, we used a local feature descriptor, namely Weighted Height Image (WHI) [19], in addition to the default Fast Point Feature Histogram (FPFH) [20]. At the third stage, we analyzed the efficiency of the hybrid approach: feature correspondence-based methods + ICP, for the problem of the co-localization of mixed-reality devices. At the final stage, we discussed the weak points of the investigated registration approaches and obtained results that could be useful for the fine-tuning of an approach for collaborative localization in real scenarios.

## 2. Related Work

This section considers the fundamental foundations of the algorithms under study and their capabilities. In the registration problem, we are given two 3D point clouds $\mathcal{A} = \{\mathbf{a}_i\}_{i=1}^{N}$ source and $\mathcal{B} = \{\mathbf{b}_i\}_{i=1}^{M}$ target point clouds with $\mathbf{a}_i, \mathbf{b}_i \in \mathbb{R}^3$.

### 2.1. Standard-ICP

This algorithm solves the $L_2$-norm registration problem for estimating rigid motion such as rotation $\mathbf{R} \in SO(3)$ and translation $\mathbf{t} \in \mathbb{R}^3$ between the source $\mathcal{A}$ and target $\mathcal{B}$ point clouds, which minimizes the objective $L_2$-error function as follows:

$$E(\mathbf{R}, \mathbf{t}) = \sum_{i=1}^{N} \|\mathbf{R}\mathbf{a}_i + \mathbf{t} - \mathbf{b}_{k*}\|^2, \tag{1}$$

where $\mathbf{a}_i$ is an $i$ point-of-source cloud and $\mathbf{b}_{k*}$ is the closest point of the target cloud to be transformed into an $\mathbf{a}_i$-point, i.e.,

$$k^* = \underset{k \in 1, \dots, M}{\operatorname{argmin}} \|\mathbf{R}\mathbf{a}_i + \mathbf{t} - \mathbf{b}_k\|. \tag{2}$$



The defined $L_2$-error is non-convex because the constraints are non-convex. The standard ICP algorithm solves this problem iteratively, alternating between estimating the transformation by (1) and finding the closest-point correspondences by (2). However, this solution guarantees convergence to the local minimum only [21].

### 2.2. Go-ICP

To find a global solution, Morrison et al. used the BnB (Branch-and-Bound) [22] algorithm, which they extended to search 3D motion $SE(3) = SO(3) \times \mathbb{R}^3$. The authors applied the domain parameterization that compactly represents the 3D rotation search space as a solid radius-$\pi$ ball in $\mathbb{R}^3$. For the translation part, the domain is represented as a bounded cube $[-\xi, \xi]$ (Figure 1).

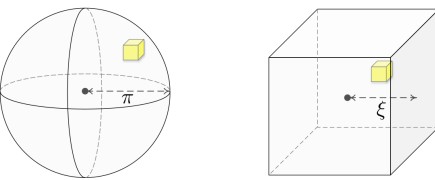

(**a**) Rotation domain          (**b**) Translation domain

**Figure 1.** (**a**) The rotation space $SO(3)$ is parameterized in a solid radius-$\pi$ ball. (**b**) The translation solution is supposed to lie within the cube with parameters $[-\xi, \xi]^3$ [23].

Yang et al. bounded the $L_2$-norm error function [23]. Domain parameterization and bounding functions allow the application of the BnB search to a problem (1). To sum up, the Go-ICP method presents the integration of two main processes: global BnB search and the local ICP search processes, which can help each other reach the global minimum of the objective function (1), as seen in Figure 2.

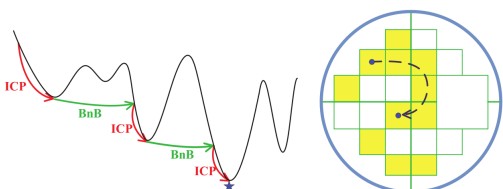

**Figure 2.** Collaboration of BnB and ICP [23].

### 2.3. Bayesian-ICP

To obtain the ICP method that can estimate a pose distribution, Bayesian-ICP combines ideas from stochastic gradient descent-ICP (SGD-ICP) [24] and stochastic gradient Langevin dynamics (SGLD) [25]. SGD-ICP uses stochastic gradient descent to solve the ICP optimization problem (1). For each iteration, small mini-batches $\mathcal{M}_k$ are formed from the source cloud iteration instead of the full point cloud. Therefore, these are associated with the closest points in the target cloud, such as in standard ICP. As such, SGD-ICP defines an update rule for six transformation parameters $\vartheta$ as follows:

$$\vartheta_{k+1} = \vartheta_k - \alpha A \vec{g}(\vartheta_k, \mathcal{M}_k), \tag{3}$$

where $\alpha$ is the learning rate, $A \in \mathbb{R}^{6 \times 6}$ acts as a pre-conditioner, and $\vec{g}$ is a gradient of the error objective function (1). The SGLD idea is to add the right amount of noise to the SGD optimization results in each iteration, which allows converging towards samples from the true posterior distribution. Therefore, applying the SGLD idea to SGD-ICP, the general SGD-ICP update rule (3) is modified by adding Gaussian noise $\eta_k \sim \mathcal{N}(0, A\alpha)$ and prior

$p(\vartheta)$ over the transformation parameters $\theta$, so the general update rule for Bayesian-ICP becomes the following:

$$\vartheta_{k+1} = \vartheta k - \frac{\alpha}{2} A\Big(-\nabla \log p(\vartheta_k) + N\vec{g}(\vartheta_k, \mathcal{M}_k)\Big) + \eta_k,$$ (4)

where $N$ is the size of the point cloud, and $\nabla \log p(\vartheta_k)$ is a gradient for prior distribution.

### 2.4. Fast Point Feature Histograms

The two registration methods are based on building correspondences due to a feature descriptor such as Fast Point Feature Histograms. Fast Point Feature Histograms [20] is a 33-dimensional local feature descriptor that describes the local geometry of space around a point in a 3D point cloud. This descriptor represents a simplified version of Point Feature Histograms (PFH), but it keeps the discriminative power of the PFH and can be calculated within milliseconds [26] due to the algorithm having a computational complexity of $O(k)$ compared with $O(k^2)$ for PFH. The FPFH feature calculation of point $a$ is divided into two steps. In the first step, its $k$ neighbors Simplified Point Feature Histogram (SPFH) based on PFH was calculated for each point. In the second step, the final histogram of $a$ is calculated as follows:

$$FPFH(a) = SPFH(a) + \frac{1}{k} \sum_{i=1}^{k} \frac{SPFH(a_i)}{\omega_i}$$ (5)

where $\omega_i$ is a weight representing the distance between point $a$ and a neighbor point $a_i$.

### 2.5. Weighted Height Image Descriptor

Weighted Height Image descriptor [19] (WHI) is a compact 3D local feature descriptor describing the 3D local shape in the point cloud. When FPFH is classified as an algorithm based on rotation-invariant metrics (RIM), the WHI feature descriptor is based on the local reference frame (LRF). LRF-based descriptors have clear advantages compared with rotation-invariant metrics. Firstly, descriptors estimate a rotation-invariant local frame (LRF), which is more repeatable and robust to occlusions and clutter. Secondly, using LRF simplifies information coding because rotation invariance is not needed, and allows saving the original information about the point cloud. The WHI descriptor combines the LRF and the 2D image representation through the weighted height image. The general calculation scheme is shown in Figure 3.

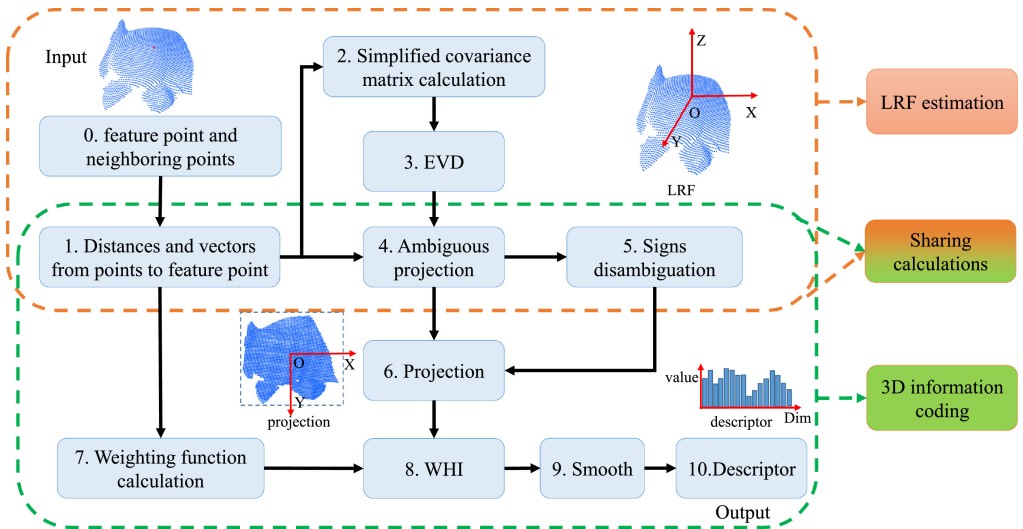

**Figure 3.** Flowchart of the WHI descriptor calculation. EVD is an eigenvalue decomposition [19].

### 2.6. Fast Global Registration

Fast Global Registration (FGR) is a registration method based on correspondences. Figure 4 shows the main module flowchart for FGR.

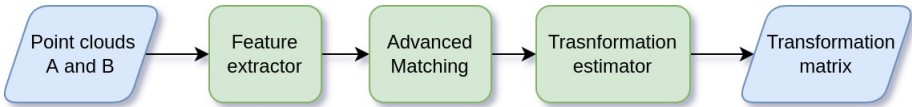

**Figure 4.** Flowchart of Fast Global Registration modules.

For each point **a** in a source cloud $\mathcal{A}$ and each point **b** in a target point cloud $\mathcal{B}$, feature points are extracted. Then, $\mathbf{F}(\mathcal{B})$ is a set of source feature points and $\mathbf{F}(\mathcal{A})$ is a set of target feature points. FGR takes the input correspondences between the source point cloud and target point cloud using feature points. Advanced matching [27] is the method that is used to build these correspondences as well as to prune the partially incorrect pairs among them. Advanced matching consists of three steps (Figure 5).

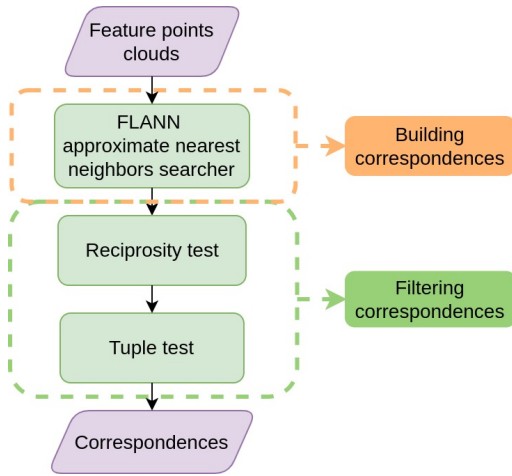

**Figure 5.** Flowchart of advanced matching.

In Step 1, set $\mathcal{K}_I$ pairs, the points of which are built by computing the nearest neighbors between feature points from $\mathbf{F}(\mathcal{A})$ to $\mathbf{F}(\mathcal{B})$ and vice versa. The Fast Library for Approximate Nearest Neighbors (FLANN) [28] kd-tree based algorithm is used to search for the nearest neighbor in multidimensional feature space. In Steps 2 and 3, a reciprocity test is applied on $\mathcal{K}_I$ to obtain $\mathcal{K}_{II}$ and a tuple test is applied on $\mathcal{K}_{II}$ to obtain the $\mathcal{K}_{III}$ set to prune correspondences [27]. Thus, to find a transformation matrix that aligns a two-point cloud, the optimization problem is solved with the following objective function:

$$E(\mathbf{T}) = \sum_{(\mathbf{b},\mathbf{a}) \in \mathcal{K}_{III}} \rho(\|\mathbf{b} - \mathbf{Ta}\|), \tag{6}$$

where $\rho$ is the penalty term. This function is important since a well-chosen penalty enables the rapid validation and pruning of bad correspondences without re-computing during optimization, as in the Standard-ICP algorithm. Zhou Q. Y., Park J., and Koltun V. used the Geman–McClure estimator penalty:

$$\rho = \frac{\mu x^2}{\mu + x^2}, \tag{7}$$

where $x$ is $\|\mathbf{b} - \mathbf{Ta}\|$ and $\mu$ is the division factor that controls the form of the objective function (6) as well as which and how many correspondences will participate in the optimiza-

tion. The optimization problem (6) cannot be solved directly. Therefore, the authors used Black–Rangarajian duality [29] which allows one to define the following objective function:

$$E(\mathbf{T}, \mathbb{L}) = \sum_{(\mathbf{b},\mathbf{a}) \in \mathcal{K}_{III}} l_{\mathbf{b},\mathbf{a}} \|\mathbf{b} - \mathbf{Ta}\|^2 + \\ + \sum_{(\mathbf{b},\mathbf{a}) \in \mathcal{K}_{III}} \mu(\sqrt{l_{\mathbf{b},\mathbf{a}}} - 1)^2, \tag{8}$$

where $\mathbb{L} = \{l_{\mathbf{b},\mathbf{a}}\}$ is a line process over the correspondences. To minimize $E(\mathbf{T}, \mathbb{L})$, the partial derivative $\partial E / \partial l_{\mathbf{b},\mathbf{a}}$ should equal zero. Thus, $l_{\mathbf{b},\mathbf{a}}$ equals the following:

$$l_{\mathbf{b},\mathbf{a}} = \left( \frac{\mu}{\mu + \|\mathbf{b} - \mathbf{Ta}\|^2} \right)^2. \tag{9}$$

If we substitute $l_{\mathbf{b},\mathbf{a}}$ in Equation (8), then objective function (8) becomes (6). This means that the optimization of (8) optimizes objective (6). The benefit of optimizing the objective (8) is its extremely efficient calculation by alternating and separately optimizing between $\mathbb{L}$ and $\mathbf{T}$. In addition, optimization (8) guarantees the convergence of the objective function (6).

### 2.7. Teaser++

Teaser++ is based on correspondences as well as FGR. In Figure 6, the main module flowchart is shown for Teaser++.

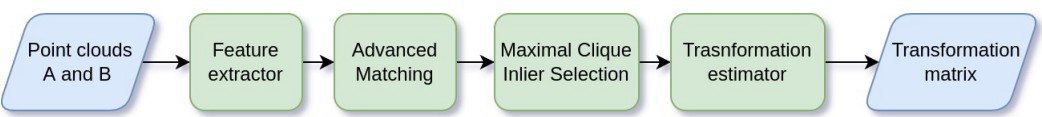

**Figure 6.** Flowchart of Teaser++ modules.

Teaser++ takes the correspondence pairs of source and target point clouds as input. Yang H., Shi J., and Carlone L. also used advance matching [27] to build correspondences the $(\mathbf{a}_i, \mathbf{b}_i), i = 1, ..., N$ using extracted feature points for each point of source $\mathcal{A} = \{\mathbf{a}_i\}_{i=1}^N$ and $\mathcal{B} = \{\mathbf{b}_i\}_{i=1}^N$ target point clouds with $\mathbf{a}_i, \mathbf{b}_i \in \mathbb{R}^3$ as in FGR. The authors also stated that point clouds contain measurement noise and therefore defined the following generative model of correspondence $(\mathbf{a}_i, \mathbf{b}_i)$:

$$\mathbf{b}_i = s\mathbf{Ra}_i + \mathbf{t} + \mathbf{o}_i + \boldsymbol{\epsilon}_i, \tag{10}$$

where $s > 0$ is the estimated scale, $\mathbf{R} \in SO(3)$ and $\mathbf{t}$ are the estimated rotation and translation, respectively, and $\mathbf{o}_i$ is an equal zero vector if the pair $(\mathbf{a}_i, \mathbf{b}_i)$ comprises inlier correspondences. However, if the pair $(\mathbf{a}_i, \mathbf{b}_i)$ is an outlier, $\mathbf{o}_i$ is just an arbitrary vector. The Truncated Least Squares (TLS) registration optimization problem is formulated as follows to find the unknown transformation between two point clouds:

$$\min_{s>0, \mathbf{R} \in SO(3) \, \mathbf{t} \in \mathbb{R}^3} \sum_{i=1}^N \left( \frac{1}{\beta_i^2} \|\mathbf{b}_i - s\mathbf{Ra}_i - \mathbf{t}\|^2, \bar{c}^2 \right), \tag{11}$$

This formulation allows the consideration of a set of correspondences which have extreme amounts of outliers, as well as inliers which have the unknown Gaussian noise but bounded and given $\beta_i \geq \|\epsilon_i\|$. The TLS only estimates solutions to the measurements that have small residuals $(1/\beta_i^2 \|\mathbf{b}_i - s\mathbf{Ra}_i - \mathbf{t}\|^2 \leq \bar{c}^2)$, so that TLS penalizes outliers and inliers with big residual errors. The optimization problem (11) is non-convex and difficult to solve directly. However, it can be solved in cascade by decoupling the estimation of scale, rotation, and translation.

For this, the authors introduced two invariant measurements. The first was the translation invariant measurement (TIM):

$$\mathbf{b}_{ij} = s\mathbf{R}\mathbf{a}_{ij} + \mathbf{o}_{ij} + \boldsymbol{\epsilon}_{ij}, \tag{12}$$

where $\mathbf{b}_{ij}, \mathbf{a}_{ij}\mathbf{o}_{ij}, \boldsymbol{\epsilon}_{ij}$ are differences in the i and j components, for example, $\mathbf{b}_{ij} = \mathbf{b}_i - \mathbf{b}_j$, etc. TIM only depends on the rotation and scale. The second was the Translation and Rotation Invariant Measurement (TRIM):

$$s_{ij} = s + o_{ij}^s + \epsilon_{ij}^s, \tag{13}$$

where $s_{ij} = \|\mathbf{b}_{ij}\|/\|\mathbf{a}_{ij}\|$, $o_{ij}^s = o_{ij}/\|\mathbf{a}_{ij}\|$, and $\epsilon_{ij}^s = \epsilon_{ij}/\|\mathbf{a}_{ij}\|$. TRIM does not depend on the rotation and translation and only depends on scale. In other words, invariant measurements enable the estimation of the scale, rotation, and translation by the three following steps:

(1)    Using TRIMs to estimate the scale $\hat{s}$;
(2)    Using TIMs and $\hat{s}$ to estimate the rotation $\hat{\mathbf{R}}$;
(3)    Using $\hat{\mathbf{R}}$ and $\hat{s}$ to estimate translation $\hat{\mathbf{t}}$ from the TLS problem (11).

As a result, the three separate optimization problems below were solved to estimate the final transformation.

1. Scale estimation:

$$\hat{s} = \underset{s}{\mathrm{argmin}} \sum_{k=1}^{K} \min\left(\frac{(s - s_k)^2}{\alpha_k^2}, \bar{c}^2\right). \tag{14}$$

where $s_k$ is equal to $\{s_{ij}\}_k, k = 1, ..., K$ invariant measurements, $\alpha_k = \{\alpha_{ij}\}_k = \{\sigma_{ij}/\|\mathbf{a}_{ij}\|\}_k$, $\sigma_{ij} = \beta_i + \beta_j$.

2. Rotation estimation:

$$\hat{\mathbf{R}} = \underset{\mathbf{R}\in SO(3)}{\mathrm{argmin}} \sum_{k=1}^{K} \min\left(\frac{\|\mathbf{b}_k - \hat{s}\mathbf{R}\mathbf{a}_k\|^2}{\sigma_k^2}, \bar{c}^2\right), \tag{15}$$

where $\sigma_k$ is equal $\{\sigma_{ij}\}_k = \{\beta_i + \beta_j\}_k$, and $\mathbf{b}_k = \{\mathbf{b}_{ij}\}_k$, $\mathbf{a}_k = \{\mathbf{a}_{ij}\}_k$, respectively.

3. Component-wise translation estimation:

$$\hat{t}_l = \underset{t_l}{\mathrm{argmin}} \sum_{k=1}^{K} \min\left(\frac{(s - [\mathbf{b}_k - \hat{s}\hat{\mathbf{R}}\mathbf{a}_k]_k)^2}{\beta_k^2}, \bar{c}^2\right), \tag{16}$$

where $t_l$ is the $l - th$ component of translation vector and $l = 1, 2, 3$. Problems (14) and (16) were solved in polynomial time by an adaptive voting algorithm [30]. The TLS rotation estimation (15) was relaxed to a tight semidefinite relaxation problem and quickly solved using the graduated non-convexity [31]. In addition, the estimation was used in a maximal clique inlier selection (MCIS) [32] after the scale estimation. This enables the removal of numerous outlier correspondences and increases the robustness to outliers.

## 3. Feature Inliers Graph Registration Approach

The Feature Inliers Graph Registration Approach (FIGRA) is a correspondence-based registration pipeline proposed in this paper. The pipeline flowchart is shown in Figure 7. The pipeline is the result of combining the best modules of Teaser++, FGR, and one new module, Hierarchical Navigable Small Worlds (HNSW). We used the HNSW algorithm instead of the Advanced Matching module. Unlike Advanced Matching, HNSW applied searching to the nearest feature points only once from the source $\mathbf{F}(\mathcal{A})$ to target $\mathbf{F}(\mathcal{B})$ to build a set of correspondences. Such a search was possible because HNSW has a significantly better recall than FLANN in Advanced Matching [33]. MSIC is a graph-based algorithm responsible for filtering correspondences, and therefore eliminates the use of additional filtering in Advanced Matching. We decided to use the transformation estimator

from the FGR method because the FGR transformation estimator required tuning only one division factor $\mu$, compared with two parameters in Teaser++.

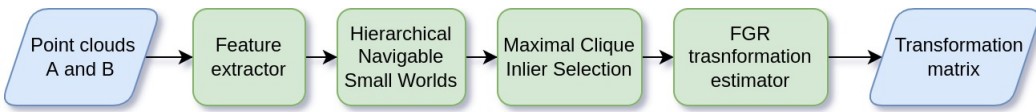

**Figure 7.** Flowchart of FIGRA modules.

*Hierarchical Navigable Small Worlds*

Hierarchical Navigable Small Worlds is a fully graph-based robust algorithm for searching approximate nearest neighbors (ANNs) in vector space created by Yu. A. Malkova and D. A. Yashunin [33]. HNSW is the natural result of the evolution of Navigable Small Worlds (NSWs). HSW is the proximity graph with long-range as well as short-range links and (poly-)logarithmic complexity search. HNSW combines the NSW and ideas of hierarchical multi-layers from Pugh's probability skip list structure which creates a performance with fast search speeds and high recall. There are later ANN graph searches, but among them, HNSW remains the most preferable in terms of the ratio of recall, building, and search time [34].

## 4. Datasets

In our work, we compared the point cloud registration algorithms for the co-localization problem of mixed-reality devices. The algorithm took the point clouds of the reconstructed environment from two different devices. The reconstructed point cloud only approximates the geometric parameters of the user's real environment, but did not describe it with high accuracy. Therefore, a dataset was required to consist of reconstructed point cloud pairs for different locations, and each pair of point clouds had to have different point distributions. This work relied on the following datasets: real datasets (dataset A and dataset B) and a synthetic dataset. Table 1 shows the key information about real datasets used for the comparative evaluation of registration algorithms.

**Table 1.** Brief descriptions of real datasets

| Code | Name | Collecting Device | Type Data | Point Cloud Density | Environments |
|---|---|---|---|---|---|
| **Dataset A** | KTH Longterm | Scitos G5 robot with an RGB-D camera sensor | Point cloud indoor environment reconstructed by different SLAM algorithms | Dense | 4 rooms, 4 corridors |
| | ICL-NUIM | RGB-D camera | | | 2 rooms |
| **Dataset B** | Indoor HoloLens | HoloLens glasses 1st and 2nd gen. | | Sparse | 3 rooms, 1 corridor |

**Dataset A. KTH Longterm and ICL-NUIM datasets.** Dataset A was compiled from open data sources. Dataset A consisted of two sub-datasets: KTH Longterm (https://strands.readthedocs.io/en/latest/datasets/kth_lt.html, accessed on 1 January 2023) and ICL-NUIM (http://redwood-data.org/indoor/dataset.html, accessed on 1 January 2023). KTH Longterm was autonomously collected by a Scitos G5 robot with an RGB-D camera on a pan-tilt. This sub-dataset contained data from eight different areas of the KTH office environment. Half of the areas were rooms, and the others were corridors. The ICL-NUIM sub-dataset was collected by RGB-D camera sensor and contained data from two rooms: living and office rooms. As a result, the first dataset contained 11 pairs of KTH Longterm

point clouds and two pairs of ICL-NUIM point clouds. In both cases, the data were not collected from mixed-reality devices.

**Dataset B. Indoor HoloLens dataset.** We collected Dataset B using two mixed-reality devices: Microsoft HoloLens 1st and 2nd gen. Each device built a mesh map of the environment. We explored one space from two devices and used the Windows Device Portal to download the Spatial mapping. Spatial mapping is the mesh, so we only took vertexes as a point cloud. Dataset B contained the sparse point clouds of four areas: three different rooms and a corridor. We obtained 20 pairs of point clouds, where each pair of point clouds was obtained from different devices.

**Synthetic dataset.** To evaluate the accuracy, we created a synthetic dataset of point clouds based on real point clouds from Dataset B (Figure 8). We selected large samples of point clouds and took different intersected parts from each large cloud. For each pair, we randomly set the ground truth transformation in the range of a $[-90, 90]$ degree for the rotation and $[-50, 50]$ cm for the translation. The basic idea of a synthetic dataset was that we knew the actual position of point cloud pairs relative to each other before registration. Hence, we could estimate the accuracy of the point cloud pair alignment.

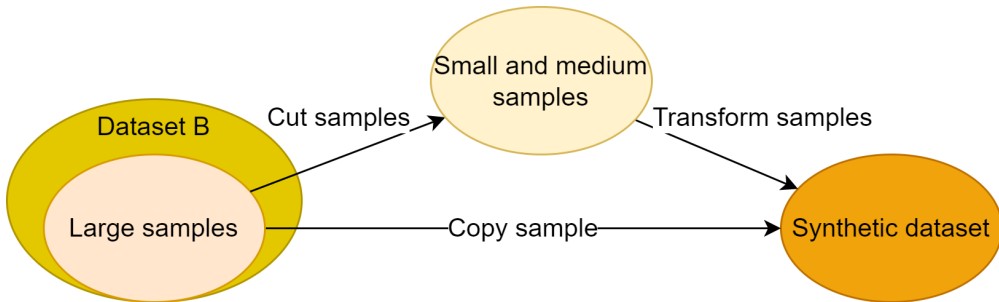

**Figure 8.** Synthetic dataset creation.

## 5. Methodology

This section describes our three-stage methodology for the comparative evaluation of the registration methods.

### 5.1. Efficiency Evaluation of Registration Algorithms

At the first stage, we evaluated the effectiveness of FIGRA and four other registration algorithms: Go-ICP, Bayesian-ICP, FGR, and Teaser++. FGR and Teaser++ were evaluated depending on Advanced Matching usage modes: On (using filtering correspondences) and Off (not using filtering correspondences). FGR, Teaser++, and FIGRA were also evaluated depending on different feature descriptors, namely the FPFH and WHI of two dimensions (16, 36), which will subsequently be referred to as WHI16 and WHI36. The used feature radius did not exceed 150 cm, with the ratio close to the recommended, which is described in the subsection below. The effectiveness assessment consisted of the average registration time and the rate of successful alignments. No ground truth information exists about the actual transformation between the pairs of point cloud origins. Thus, the alignment success for each pair was visually evaluated. We considered successful registration to be the alignment of the planes of the room geometry in the area of overlapping point clouds while the aligned planes were not visually separate planes, as shown in the example (Figure 9). All algorithms were tested on Datasets A and B.

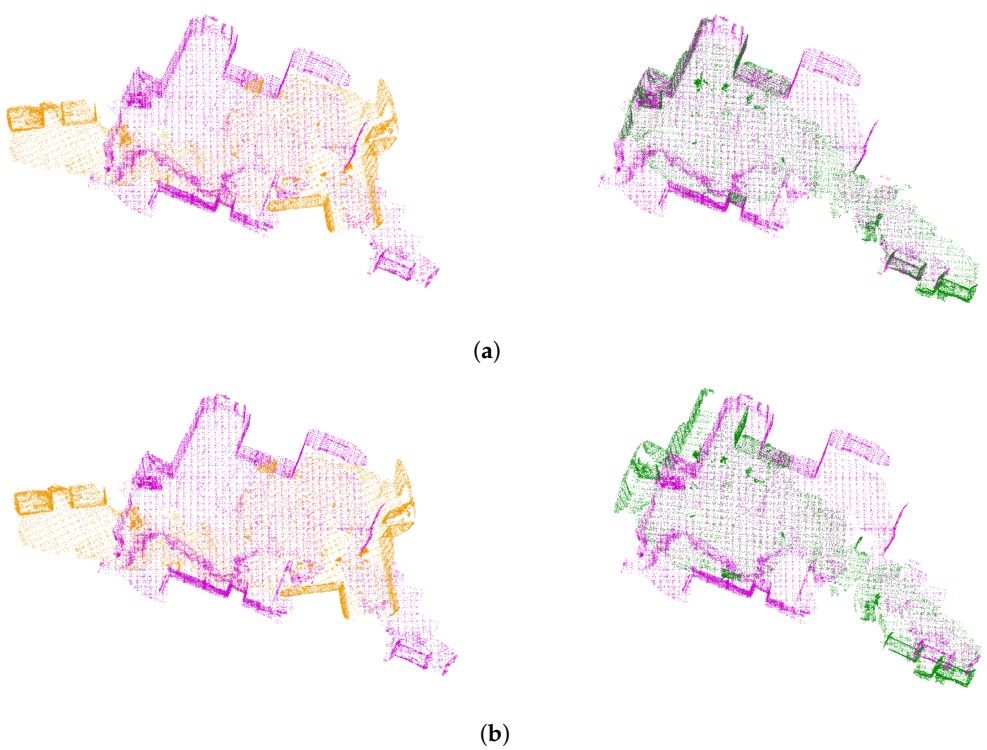

**(a)**

**(b)**

**Figure 9.** Examples of visually successful and non-successful registrations. Target point cloud—magenta; source point cloud—yellow; aligned point cloud—green. The left half of the sub-figures shows the state before registration and the right half shows a state after registration. (**a**) Successful alignment. (**b**) Non-successful alignment

*5.2. Accuracy and Runtime Analysis of Registration Methods: FGR, Teaser++, and FIGRA for Different Local Feature Descriptors*

At the second stage, we studied the efficiency of the FGR, Teaser++, and FIGRA algorithms depending on the different radii of the features FPFH, WHI16, and WHI36. We used the synthetic dataset for accuracy and runtime evaluation. To find the rotation error, we used the roll, pitch, and yaw angles calculated for the transformation matrix $T_a$ obtained by the algorithm and for the ground truth transformation matrix $T_g$:

$$\phi = \text{atan2}\,(r_{32}, r_{33}), \theta = \arcsin\,(-r_{30}),$$
$$\psi = \text{atan2}\,(r_{21}, r_{11}), \tag{17}$$

where $r_{ij}$ is the $ij$ element of the rotation part in the transformation matrix. Rotation error $R_{error}$ was defined as a summary error of roll ($\phi_{error}$), pitch ($\theta_{error}$), yaw ($\psi_{error}$) angles:

$$\phi_{error} = |\phi_g - \phi_a|, \theta_{error} = |\theta_g - \theta_a|,$$
$$\psi_{error} = |\psi_g - \psi_a|, \tag{18}$$
$$R_{error} = \phi_{error} + \theta_{error} + \psi_{error}.$$

Translation error $t_{error}$ was calculated following:

$$t_{error} = \sqrt{(x_g - x_a)^2 + (y_g - y_a)^2 + (z_g - z_a)^2}. \tag{19}$$

We also evaluated the success rate of registrations for FGR, Teaser++, and FIGRA. For the FPFH, WHI16, and WHI36 local feature descriptors, we calculated the metrics for different feature radii, but we kept the optimal ratio downsampling $(r_d)$/normal $(r_n)$/feature $(r_f)$ radius equal to $1:2:5$ for FPFH and the optimal ratio downsampling $(r_d)$/feature $(r_f)$ radius equal to $1:5$ for WHI types feature descriptors. FPFH ratio was

recommended by the authors of FGR, but we chose the WHI ratio for more reliable comparison of correspondence-based registration methods with different feature descriptors. We considered that the alignment of two synthetic point clouds is successful with the following accuracy condition:

$$R_{error} \leq 0.03 \text{ rad}; \quad t_{error} \leq 1.0 \text{ cm}. \tag{20}$$

We considered such accuracy requirements to be satisfactory for the co-localization of mixed-reality devices. The registration time was considered to be sufficient within no more than 5 s for the successful synchronization of the devices, including further update and refinement. Thus, we considered the registration to be successful if it complies with the following conditions: 100% cases satisfy the accuracy condition (20) and a runtime below 5 s.

### 5.3. Accuracy and Runtime Analysis of Hybrid Approaches

At the third stage, we evaluated the registration efficiency of the hybrid approach (correspondence-based method [FGR or Teaser++ or FIGRA] as coarse + ICP as local refinement). The hybrid approach allowed us to use the advantages of the two techniques. The first one did not require closed initialization for the point cloud registration, but the usage of downsampling limits the accuracy of the method. The second technique had a high convergence accuracy (registration moves to a local minimum, and converges globally only when close to the global minimum), but it requires the appropriate initial position of point clouds relative to each other. As such, the first technique allowed the exclusion of the disadvantages of the second, and the second method allowed excluding the disadvantages of the first one.

We used different ICP iteration numbers to evaluate the effectiveness of the hybrid approach and to discover the working range of the feature radius. We also used the previous metrics, and we added a new criterion for a more flexible evaluation: 90% of cases satisfy the accuracy condition (20) and stay within 10 seconds. Accuracy and runtime can be improved by using the hybrid approach, submodule improvements, and more powerful hardware.

All experiments were performed using Point Cloud Library (PCL) [35] on a laptop with CPU AMD Ryzen 7 4800HS.

### 6. Results

### 6.1. Efficiency Evaluation of Registration Algorithms on Real Datasets

Table 2 shows the estimated registration time and the rate of success alignments for point cloud pairs. The algorithms were based on ICP: Go-ICP and Bayesian-ICP showed very low success rates in terms of aligning with a high registration time. Feature-based algorithms (FGR, Teaser++, and FIGRA) significantly outperformed ICP-based ones in terms of both successful alignments and execution time. FIGRA had a higher registration success compared with other methods and at the same time, FIGRA had a low average registration runtime. Furthermore, the results for FIGRA showed a low correlation between the registration time and the feature dimension.

Dataset A demonstrated a lower rate of successful alignments by the algorithms compared to dataset B. This difference indicates that Dataset B had cloud pairs that were more difficult to register. Figures 10 and 11 show this difference between point clouds with samples of the point cloud registration for different datasets. In Dataset B, more than half of the point cloud pairs had a small overlap fraction: less than 50%, and more different points distribution. The registration success of a point cloud pair depended on the overlap fraction as well as on the initial degree of point cloud sparsity. The overlap fraction determined how many pairs of point clouds had common geometric parts. In other words, the greater was the overlap area of the point clouds, the easier it was for the registration algorithm to find correspondences between those clouds and match them to each other. The sparsity degree of the point clouds affected the degree of dissimilarity in point cloud distribution. The sparse clouds may have had more dissimilar point distributions than dense clouds.

**Table 2.** Efficiency result of registration methods on **dataset A** (left) and **dataset B** (right).

| Method | Feature | Advanced Matching | Average Runtime (ms) | Alignment Success (%) |
|---|---|---|---|---|
| Go-ICP | - | - | 24427 | 8 |
| Bayesian-ICP | - | - | 1647 | 54 |
| FGR | FPFH | On | 390 | 100 |
|  | WHI16 | On | 371 | 100 |
|  | WHI36 | On | 752 | 100 |
|  | FPFH | Off | 442 | 100 |
|  | WHI16 | Off | 441 | 100 |
|  | WHI36 | Off | 823 | 92 |
| Teaser++ | FPFH | On | 409 | 100 |
|  | WHI16 | On | 428 | 100 |
|  | WHI36 | On | 823 | 100 |
|  | FPFH | Off | 1847 | 100 |
|  | WHI16 | Off | 1209 | 100 |
|  | WHI36 | Off | 1897 | 100 |
| FIGRA | FPFH | - | 1359 | 100 |
|  | WHI16 | - | 613 | 100 |
|  | WHI36 | - | 687 | 100 |
| **Method** | **Feature** | **Advanced Matching** | **Average Runtime (ms)** | **Alignment Success (%)** |
| Go-ICP | - | - | 24158 | 0 |
| Bayesian-ICP | - | - | 1564 | 5 |
| FGR | FPFH | On | 219 | 68 |
|  | WHI16 | On | 223 | 53 |
|  | WHI36 | On | 419 | 79 |
|  | FPFH | Off | 259 | 42 |
|  | WHI16 | Off | 262 | 26 |
|  | WHI36 | Off | 458 | 42 |
| Teaser++ | FPFH | On | 219 | 63 |
|  | WHI16 | On | 213 | 58 |
|  | WHI36 | On | 446 | 63 |
|  | FPFH | Off | 365 | 79 |
|  | WHI16 | Off | 382 | 100 |
|  | WHI36 | Off | 641 | 100 |
| FIGRA | FPFH | - | 350 | 89 |
|  | WHI16 | - | 288 | 100 |
|  | WHI36 | - | 311 | 100 |

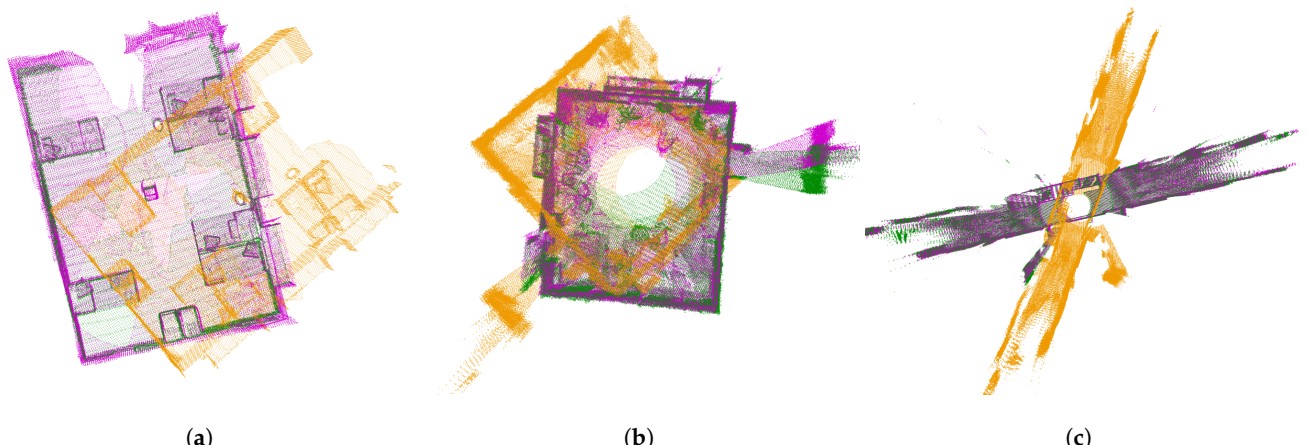

(**a**)  (**b**)  (**c**)

**Figure 10.** Registration samples on **dataset A**. Target point cloud—magenta; source point cloud—yellow; aligned point cloud—green; (**a**) Sample 1. (**b**) Sample 2. (**c**) Sample 3.

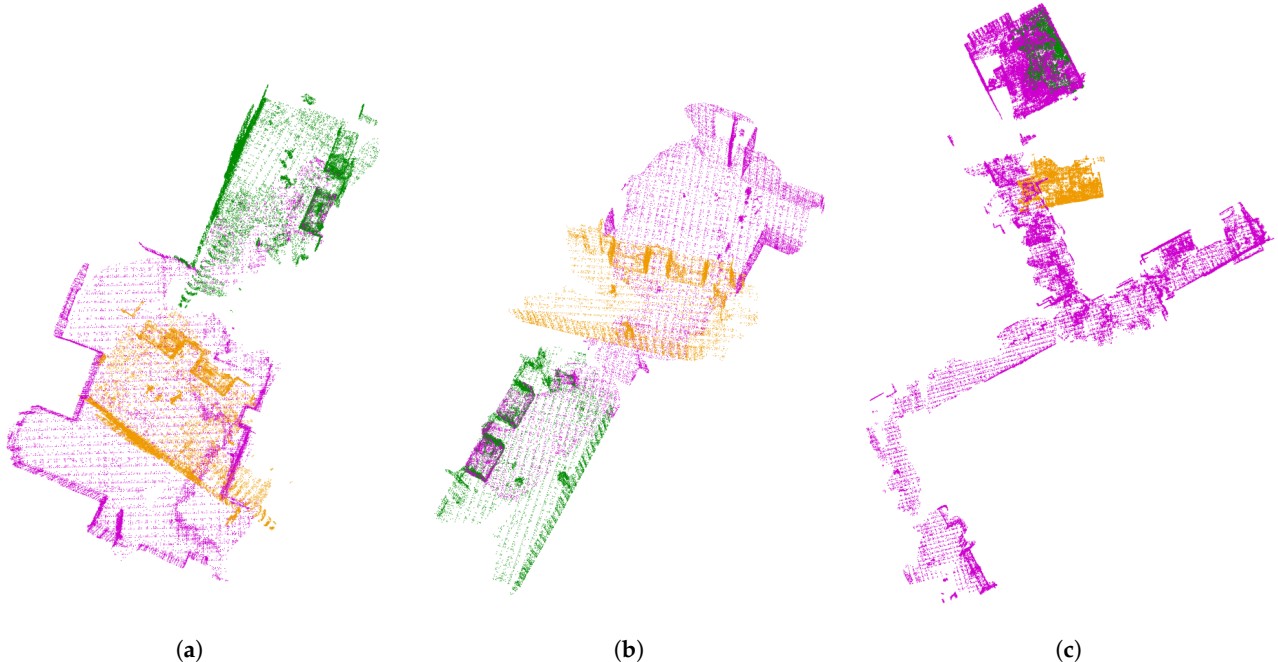

    (**a**)                  (**b**)                  (**c**)

**Figure 11.** Registration samples on **dataset B**. Target point cloud—magenta; source point cloud—yellow; aligned point cloud—green. (**a**) Sample 1. (**b**) Sample 2. (**c**) Sample 3.

When the Advanced Matching for the FGR algorithm was in the Off mode (turned off filtering correspondences), the probability of successful point cloud alignment tended to decrease. This tendency indicated that the FGR estimator could not cope with outliers without additional filtering outliers. On the contrary, for Teaser++, when Advanced Matching was in the Off mode, the percentage of successful point cloud alignments increased and reached 100% with local feature descriptors WHI16 and WHI36. This increase might have happened because the pruning correspondence part of Advanced Matching rejected not only the incorrect correspondences but also part of the correct ones, and the MCIS submodule of Teaser++ effectively selects inliers.

### 6.2. Accuracy and Runtime Analysis of FGR, Teaser++, and FIGRA for Different Feature Descriptors

Figures 12–14 show the accuracy and runtime results of FGR, Teaser++, and FIGRA with different local feature descriptors. The rotation and translation accuracy of the methods with the local feature descriptors WHI16 and WHI36 exceeded the accuracy of the methods with FPFH, while the runtime of methods with feature WHI16 was the shortest for a feature radius of more than 50 cm. In Figures 12–14, one can notice that point cloud registration translation accuracy was less than the downsampling level for feature radius 150 cm. Voxel grid downsampling allowed saving the surface structure, since a centroid point was calculated for each voxel. Hence, on the one hand, the small voxel size allowed the slight reduction in the number of calculations without losing information about the surface geometry. On the other hand, we significantly accelerated the calculations with a large feature radius.

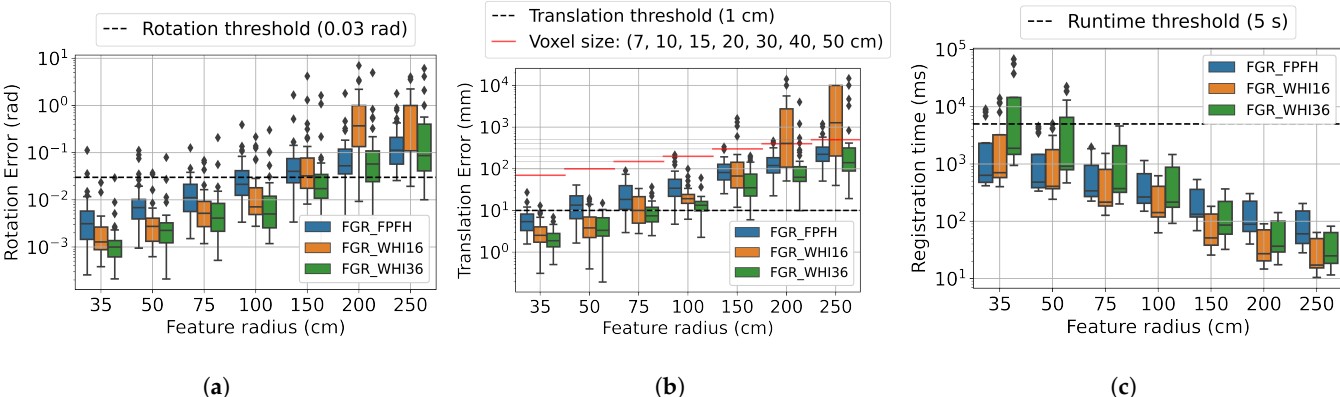

(**a**)                                                    (**b**)                                                    (**c**)

**Figure 12.** FGR_FPFH vs. FGR_WHI16 vs. FGR_WHI36. (**a**) Rotation estimation. (**b**) Translation estimation. (**c**) Runtime estimation.

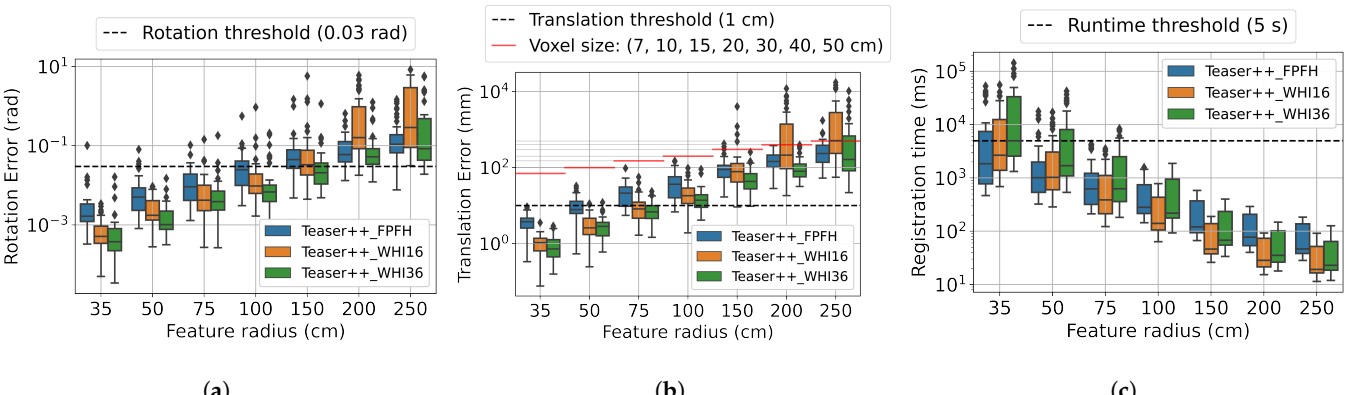

(**a**)                                                    (**b**)                                                    (**c**)

**Figure 13.** Teaser++_FPFH vs. Teaser++_WHI16 vs. Teaser++_WHI36. (**a**) Rotation estimation. (**b**) Translation estimation. (**c**) Runtime estimation.

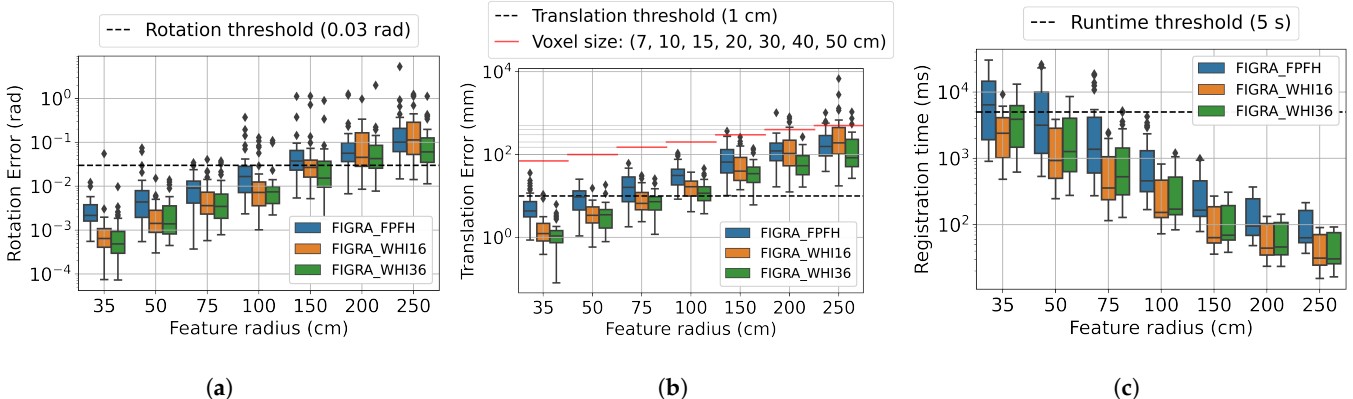

(**a**)                                                    (**b**)                                                    (**c**)

**Figure 14.** FIGRA_FPFH vs. FIGRA_WHI16 vs. FIGRA_WHI36. (**a**) Rotation estimation. (**b**) Translation estimation. (**c**) Runtime estimation.

Therefore, the optimal range of the feature radius satisfied some accuracy and runtime requirements of real applications, which was a compromise between accuracy and runtime.

*6.3. Accuracy and Runtime Analysis of Hybrid Approaches: FGR, Teaser++, and FIGRA with ICP*

We used FGR, Teaser++, and FIGRA with ICP as a hybrid approach. Table 3 shows the summary evaluation of FGR, Teaser++, and FIGRA for different types and parameters of local feature descriptors and ICP iterations. In addition, the table provides the radius ranges of the feature descriptors that correspond to the defined accuracy and execution time criteria. Table 3 can be useful in determining the optimal feature radius range needed to solve the collaborative localization problem.

**Table 3.** Results for FGR, Teaser++, and FIGRA on the **synthetic dataset**. Table shows the algorithms' performance with different configurations, namely feature descriptors and ICP iterations.

| Method | Feature | ICP | Feature Radius (cm) | | | | | | |
|---|---|---|---|---|---|---|---|---|---|
| | | | 35 | 50 | 75 | 100 | 150 | 200 | 250 |
| FGR | FPFH | - | | | | | | | |
| | | 1 | | | | | | | |
| | | 10 | | | | | | | |
| | | 100 | | | | | | | |
| | WHI16 | - | | | | | | | |
| | | 1 | | | | | | | |
| | | 10 | | | | | | | |
| | | 100 | | | | | | | |
| | WHI36 | - | | | | | | | |
| | | 1 | | | | | | | |
| | | 10 | | | | | | | |
| | | 100 | | | | | | | |

| Method | Feature | ICP | Feature Radius (cm) | | | | | | |
|---|---|---|---|---|---|---|---|---|---|
| | | | 35 | 50 | 75 | 100 | 150 | 200 | 250 |
| Teaser++ | FPFH | - | | | | | | | |
| | | 1 | | | | | | | |
| | | 10 | | | | | | | |
| | | 100 | | | | | | | |
| | WHI16 | - | | | | | | | |
| | | 1 | | | | | | | |
| | | 10 | | | | | | | |
| | | 100 | | | | | | | |
| | WHI36 | - | | | | | | | |
| | | 1 | | | | | | | |
| | | 10 | | | | | | | |
| | | 100 | | | | | | | |

| Method | Feature | ICP | Feature Radius (cm) | | | | | | |
|---|---|---|---|---|---|---|---|---|---|
| | | | 35 | 50 | 75 | 100 | 150 | 200 | 250 |
| FIGRA | FPFH | - | | | | | | | |
| | | 1 | | | | | | | |
| | | 10 | | | | | | | |
| | | 100 | | | | | | | |
| | WHI16 | - | | | | | | | |
| | | 1 | | | | | | | |
| | | 10 | | | | | | | |
| | | 100 | | | | | | | |
| | WHI36 | - | | | | | | | |
| | | 1 | | | | | | | |
| | | 10 | | | | | | | |
| | | 100 | | | | | | | |

Legend:

- (green upper triangle) : configurations with 100% results satisfied accuracy below 0.03 rad and 1 cm
- (yellow upper triangle) : configurations with more than 90% results satisfied accuracy below 0.03 rad and 1 cm
- (white upper triangle) : configurations with less than 90% results satisfied accuracy below 0.03 rad and 1 cm
- (green lower triangle) : configurations with 100% of execution runtime below 5 seconds
- (yellow lower triangle) : configurations with 100% of execution runtime below 10 second
- (white lower triangle) : configurations with less than 100% of execution runtime below 10 second

After testing, we noticed an increased registration accuracy for all feature radii not exceeding 150 cm. However, we were not satisfied with the poor accuracy of hybrid FGR and Teaser++ methods for the WHI16 feature radius of over 150 cm. This accuracy created faulty initialization for ICP, and as a result, ICP fell to a local minimum. Importantly, the hybrid approach for the FPFH feature radius of over 150 cm allowed the ICP to converge to a global minimum. This allowance indicated that the accuracy of FGR and Teaser++ for the FPFH feature radius of more than 150 cm was greater than for the WHI feature type. WHI for a large feature radius might have partially lost its descriptiveness, unlike the FPFH feature.

The results of the FIGRA evaluation on the synthetic dataset showed an almost complete satisfaction of the accuracy condition for all feature descriptors and feature radii. Based on the results with the synthetic dataset, the best feature descriptor to be used in a real application could not be reliably determined. However, the real-data results showed the superiority of the WHI feature over FPFH. Therefore, we recommend using the WHI feature in the feature radius range below 150 cm, and used FPFH for the radius extending 150 cm.

## 7. Discussion

The results show that ICP-based methods were not suitable for solving the co-localization problem, unlike feature-based algorithms. However, the use of ICP to refine the feature-based methods showed a good result. Among feature-based algorithms, the proposed FIGRA pipeline excelled in terms of the runtime and success of registration on real data. The main limitation of all registration algorithms under this study is the dependence on the performance on the SLAM algorithms. If the AR/MR device had an error relative to

the map, then this error was added to the co-localization error. The spatial map was a derivative of SLAM algorithms onboard devices and was updated in a few seconds, so the co-localization procedure would be called even less frequently. Moreover, the sparse density of spatial maps and the dissimilarity in mesh sizes created additional complexity for point cloud registration. Furthermore, this approach did not comply with the principles of confidentiality. However, if a user wanted to manage a map for content location, then they would have to save its original form with private information.

The analysis of FGR and Teaser++ methods on real datasets revealed their weakness in the advanced matching module. As the results show, the collaborative localization of MR devices or registration success depends on the quality of building and filtering of correspondences. Filtering correspondences is more important for point cloud maps with a small overlap because, in this case, the number of outliers increases. Advanced Matching used the ineffective method FLANN of the nearest search neighbor in the kd-tree with low recall for a high dimension space. To compensate for this drawback, Advanced Matching used a cross-search that significantly increases the runtime. Therefore, we replaced this module with HNSW, which had a much better recall than FLANN in Advanced Matching. Hence, it was allowed to increase the quality of co-localization in terms of accuracy and registration success probability. Moreover, the module HNSW makes it possible to use the feature descriptors of higher dimensions without large losses in the registration time. Furthermore, the results show that using the WHI feature descriptor instead of the default FPFH descriptor allowed for increasing the probability of successful registration.

## 8. Conclusions

In this paper, we proposed the FIGRA pipeline to the mixed-reality cross-device localization and showed its performance and limitations. Furthermore, we compared the efficiency of the five following point cloud registration methods: Go-ICP, Bayesian-ICP, FGR, Teaser++, and FIGRA on real point clouds of rooms obtained by Microsoft HoloLens (1st and 2nd gen) MR devices. The feature correspondence-based methods: FGR, Teaser++, and FIGRA showed a millisecond runtime efficiency and high probability of successful alignments compared with the following ICP-based methods: Go-ICP, Bayesian-ICP. Among the proposed feature-based methods, the FIGRA pipeline was better in terms of runtime and successful registration on real data. Hence, using FIGRA is preferable for the collaborative localization of mixed-reality devices. We also tested a new WHI feature descriptor for the point cloud registration method. We tested a hybrid approach on synthetic data and provided the table with different algorithm parameters and performance for co-localization MR devices. For the co-localization of MR devices in a real scenario, we recommend using the WHI feature descriptor with a feature radius of less than 150 cm, as it was more robust to interference and descriptive on real data compared to FPFH.

In future works, we would like to extend the approach for the co-localization of the multi-robot system and multi mixed-reality device in one space. We will investigate the methods for the localization of MR devices in large pre-built and labeled maps.

**Author Contributions:** Conceptualization, M.O. and A.K.; Methodology, A.O. and M.O.; Software, A.O. and M.O.; Validation, A.O. and M.O.; Formal analysis, M.O.; Investigation, A.O.; Resources, M.O.; Data curation, A.O.; Writing—original draft, A.O. and M.O.; Writing—review & editing, A.K.; Visualization, A.O. and M.O.; Supervision, A.K.; Project administration, M.O.; Funding acquisition, A.K. All authors have read and agreed to the published version of the manuscript.

**Funding:** This research was funded by the Ministry of Education and National Technological Initiative in the frame of creation Center for Technologies in Robotics and Mechatronics Components (ISC 0000000007518P240002).

**Data Availability Statement:** The results were obtained on the publicly available datasets https://strands.readthedocs.io/en/latest/datasets/kth_lt.html, and http://redwood-data.org/indoor/dataset.html (accessed on 1 January 2023).

**Conflicts of Interest:** The authors declare no conflict of interest.

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
