# Peer review of "Comparison of Point Cloud Registration Algorithms for Mixed-Reality Cross-Device Global Localization"

_information, doi:10.3390/info14030149_

Round 1

Reviewer 1 Report

·         Major Comments

o   Syntax and grammatical errors and the overall linguistic quality of the submitted manuscript is such, that it is hard for the reader to follow.

o   In the related work section, the point cloud registration methods are presented in big detail in contrast to a brief description of the proposed method in section 3. For example, the HSNW (Hierarchical Navigable Small Worlds) algorithm that is proposed as the main difference in the proposed FIGRA pipeline is not presented in detail like the other algorithms in section 2, while one could argue it has a superior significance for the submitted paper.

·         Minor Comments

o   In Figure 3, there is no description for the EVD abbreviation.

o   In section 5.2, it is mentioned that due to no ground truth information the evaluation is visual. Such a visual evaluation is largely subjective and the authors are encouraged to provide some clarification over this evaluation other than the Figure 9 that is provided.

o   The Table 3 legend descriptions contain are hard to comprehend (e.g. the meaning of  “more 90% cases … “, “less 90% cases …”)

o   While WHI (Weighted Height Image) descriptor is presented in the related work sections, WHI16 and WHI36 descriptors are mentioned in sections 5 and 6, with no clarification regarding the numbering. In addition, in the conclusion it is stated that “we tested a new WHI feature descriptor… “, while there is no mention of this in the rest of the manuscript.   

Author Response

Dear reviewer,

Please find in the attached file a detailed response to your comments and correction summary. We also did proofreading for the paper with a native speaker and a non-involved colleague

With best regards,

Alexander Osipov, Mikhail Ostanin and Alexandr Klimchik

Reviewer 2 Report

The proposed algorithm (called FIGRA by the authors) is a small variation of the Teaser++ approach. For such a variation the paper seems to be too long having 16 pages in total. Furthermore, the authors should make their contribution clearer:  Is the contribution the new method as described in the abstract and introduction or is the main contribution the analysis and comparison of the different algorithms (FGR, Teaser++, and FIGRA) as one could expect from the title?

In some cases, the content of related work is not properly explained. The authors give [15] as an example of visual tracking, but [15] presents three different methods for tracking. Only one of which uses a visual marker (an ArUco marker to be precise). The other two methods work differently. The authors write that this marker is a QR code, but it is a fiducial marker, a visual pattern particularly suited for pose estimation. QR codes, on the other hand, are two-dimensional barcodes that contain information.

The clarity can be improved:

E.g. in Figure 10: what are the different samples? This is not explained in the caption or neither in the text.

Why is the alignment in Figure 10 b) so bad? Looks like a failure case.

Author Response

(The authors gave the same response as above.)

Round 2

Reviewer 1 Report

The paper can be published in its current form.